# Long-Term Outcomes (10 Years) of Sacrospinous Ligament Fixation for Pelvic Organ Prolapse Repair

**DOI:** 10.3390/healthcare12161611

**Published:** 2024-08-13

**Authors:** Annalisa Vigna, Marta Barba, Matteo Frigerio

**Affiliations:** 1Department of Gynecology, IRCCS Policlinico San Martino, University of Genova, 16126 Genova, Italy; annalisa.vigna.92@gmail.com; 2Department of Gynecology, IRCCS San Gerardo dei Tintori, University of Milano-Bicocca, 20900 Monza, Italy; m.barba8792@gmail.com

**Keywords:** post-hysterectomy vault prolapse, native-tissue repair, high uterosacral ligament suspension, sacrospinous ligament fixation, transvaginal repair, pelvic organ prolapse

## Abstract

Vaginal vault prolapse is one of the main reasons for reoperation in patients with pelvic organ prolapse. Effective correction of the vaginal apex is essential for lasting repair for these women. Apical suspension of the sacrospinous ligament is probably one of the main vaginal treatments still offered to patients today. We proposed an evaluation of the functional and anatomical results of long-term sacrospinous ligament fixation. Objective: The purpose of this study was to evaluate the 10-year results of sacrospinous ligament suspension as primary repair for apical prolapse and to evaluate long-term side effects. Materials and Methods: A retrospective study analyzed 10-year follow-up after prolapse repair using sacrospinous ligament suspension. A subjective recurrence was identified as the postoperative occurrence of swelling symptoms based on a particular item on the Italian Prolapse Quality of Life (P-QoL) questionnaire. An objective recurrence was defined as a postoperative decline to stage II or below in any compartment based on the POP-Q system or the requirement for additional surgery. The assessment of postoperative subjective satisfaction was conducted using the Patient Global Impression of Improvement (PGI-I) score. Results: In total, 40 patients underwent sacrospinous ligament fixation. Objective recurrence was remarkably high, as it was observed in 17 (56.7%) patients. Subjective recurrence was reported by ten (33.3%) women, and reintervention occurred in two (6.7%) of patients. From the point of view of quality of life, according to the PGI-I, twenty-three (76.7%) patients described some degree of improvement after surgery, four (13.3%) described their status as unmodified, and three (10%) reported some form of worsening after primary treatment. Conclusions: Transvaginal repair with sacrospinous fixation is a long-lasting option for prolapse repair, with improvement in every POP-q parameter. Some degree of anterior recurrence, recurrence of symptoms with swelling, or an overall worsening of quality of life after surgery is possible.

## 1. Introduction

Pelvic organ prolapse (POP) is a common disorder that causes a significant impact on quality of life and significant costs to the healthcare system [1,2]. A considerable increase in cases of pelvic organ prolapse has been predicted in the next twenty years due to the aging of the population [3]. From an anatomical point of view, it is possible to identify the prolapse of the anterior compartment (bladder, urethra), of the apical compartment (uterus, vaginal vault), or of the posterior compartment (rectum, intestine). Apical prolapse is the dominant component [4,5]. A woman’s quality of life is significantly impacted by pelvic organ prolapse, which can cause physical discomfort, psychological issues, sexual difficulties, and limitations at work and in social situations [6]. Surgery, if indicated, is generally the definitive solution [7]. It is critical to be able to restore the proper anatomy of the vagina and address or enhance the function of these areas in order to lower the chance of relapse [7,8].

There are several treatment options. Simple observation, vaginal pessaries, and rehabilitation of the pelvic floor muscles represent conservative management. Surgical management includes many approaches to treat POP, including as robotic, open abdominal, laparoscopic, and vaginal operations; it also entails determining whether uterine preservation is required and whether to use native-tissue restoration or transvaginal mesh [9]. The increased life expectancy and improved quality of life of postmenopausal women with pelvic organ prolapse (POP) have made repair of the apical support an essential surgical step [10]. As mentioned above, the main surgical techniques for uterine prolapse include vaginal, laparoscopic, and abdominal approaches. With this FDA categorization, native-tissue restoration in laparoscopic and vaginal surgery has gained importance [11].

Surgery by the abdominal route is usually performed in the form of sacrocolpopexy. Trans-abdominal sacrocolpopexy, both laparoscopic and robotic, has many advantages compared to the open approach: postoperative infections and pain after surgery are minor complications if a minimally invasive approach is preferred, and recovery times are therefore faster than for open surgery [12]. The advantage of minimally invasive approaches is that optimal visualization of the pelvic anatomy and presacral space is guaranteed. On the other hand, the main limitation of this approach is the need for extensive dissection, with the possibility of having significant bleeding from the presacral blood vessels. Furthermore, abdominal surgery is known to have higher morbidity rates and longer convalescence times than vaginal or laparoscopic surgery [13,14]. In vaginal surgery, native-tissue regeneration eliminates prosthesis-related problems such as infection and mesh erosion [15,16]. Therefore, the modality of repair of vaginal apical POP with and without uterine preservation, with or without the use of prosthetic material, is the cornerstone of a long-standing discussion in the field of reconstructive pelvic medicine and surgery [17]. To date, the main surgical techniques are fixation of the sacrospinous ligament (SSL) and suspension of the uterosacral ligament (USL). The primary drawback of these methods is that native-tissue operations may increase the chance of recurrence over the medium and long terms [18].

Fixation of the uterosacral ligament (USL) is a reliable, long-lasting method for correcting the central compartment syndrome [19,20]. Specifically, other studies have shown that, among the structures that support the uterus and the pelvic floor (pubourethral ligaments, uterosacral/cardinal ligament complex, and arcus tendineus fasciae pelvis) [21], the uterosacral/cardinal ligament complex is the cornerstone [22]. Damage to this complex structure can cause symptoms such as vaginal and uterine prolapse. Additionally, uterosacral vault suspension can typically keep the vaginal axis in the proper orientation, which may lower the likelihood that prolapse will repeat in other vaginal compartments, especially the anterior one [23].

Sacrospinous ligament (SSL) fixation is one of the most used transvaginal techniques for vaginal vault suspension. This procedure, which was reported by Sederl, entails suspending the vaginal apex from the sacrospinous ligaments bilaterally or unilaterally [24]. The sacrospinous ligament can be used to surgically repair apical prolapse, resulting in less invasive and more effective treatment [25]. To date, several described vaginal surgical approaches have been identified to access the SSL and allow suspension of the vaginal vault [26]. Conventionally, this is obtained through a posterior approach that, through an incision in the posterior vaginal wall, allows dissection through the pararectal space [27]. In order to properly place the sutures with this approach, a posterior colpotomy and dissection of the tissues up to the ischial spine are necessary. It appears that using this method can lower the danger of bleeding and infections related to the previously mentioned strategy. Hemorrhage is rare because extensive dissection is avoided, and mesh complications are entirely avoided due to the exclusive use of sutures [12,13,14,15,16,17,18,19,20,21,22,23,24,25,26,27,28].

The anterior approach is less extensively described and studied than the traditional posterior approach. This technique is particularly useful in women who do not have posterior prolapse and who do not require a posterior vaginal incision and dissection [29,30,31]. Again, sacrospinous ligament suspension, although shown to be successful in suspending the prolapsed vaginal vault, has unfortunately been associated with high postoperative rates of cystocele formation [32,33]. Many studies have reported that recurrent anterior vaginal wall prolapse is the most common complication of SSLF [34,35]. Fixation of the sacrospinous ligament causes an axial tilt of the vagina towards the hanging side and posteriorly, increasing the risk of anterior pelvic defects [36] compared to uterosacral ligament suspensions.

Moreover, while long-term efficacy data are available for USL suspension, this is not true for SSF, whose long-term outcomes have not been evaluated yet. Consequently, with this study, we wanted to demonstrate that, although the vaginal approach is among the best surgical approaches, sacrospinous ligament suspension is associated with high postoperative rates of anterior prolapse [37].

## 2. Materials and Methods

Between October 2008 and December 2012, patients who underwent native-tissue repair through SSL fixation for primary utero-vaginal prolapse in a single center were retrospectively analyzed. Preoperative evaluation included a medical interview to assess complete medical history and obstetric history. The presence of urinary, sexual, and bowel symptoms was defined according to the standardization of terminology of the International Urogynecology Association and the International Continence Society [38]. POP was staged using the Pelvic Organ Prolapse Quantification System (POP-Q) after a urogynecological examination [39].

The POP-Q system is based on precise measurements taken at predetermined vaginal wall midline sites. The hymenal ring is still the measurement’s fixed reference point. The POP-Q system has six designated measuring points, namely, Aa, Ba, C, D, Ap, and Bp, and three additional landmarks, namely, TVL, GH, and PB. The hymen plane is specified as zero (0). Each measurement is made in centimeters above or proximal to the hymen (negative number) or centimeters below or distal to the hymen (positive number). Because the hymen is more clearly identified than the introitus, it was chosen as the reference point. Aa, Ba, and C are the three reference points located anteriorly, whereas Ap, Bp, and D are the three points located posteriorly. Points Aa and Ap are located 3 cm anteriorly and posteriorly, respectively, above and proximal to the hymenal ring. The lowest points of the prolapse between the vaginal apex and points Aa anteriorly or Ap posteriorly are referred to as points Ba and Bp. Point C (cervix) is the apex anteriorly, while point D (Douglas pouch) is the posterior apex. Point D is skipped in women who have had a hysterectomy, and point C is the vaginal cuff. In addition, three measurements are made: the perineal body (pb) from the posterior aspect of the genital hiatus to the mid-anal opening, the genital hiatus (gh) from the middle of the urethral meatus to the posterior hymenal ring, and the vaginal length at rest [40].

Using a posterior approach, two size 0 absorbable monofilament sutures per side were used to fix the sacrospinous ligament (Figure 1) [41]. When and as needed, additional surgical procedures were carried out, such as anterior and/or posterior repair. To be more precise, the apex of the duplicated fascia was included with suspension sutures, and the midline fascia was plicated using interrupted nonabsorbable sutures from the level of the bladder neck to the anterior vaginal wall apex.

Surgeons with competence in pelvic floor surgery carried out each procedure. Following surgery, patients were checked on annually for the first year. Patients who had not had a visit in the previous year were contacted by phone and given an appointment time. A thorough urogenital examination and a clinical interview were part of the follow-up visits. The subjective recurrence of postoperative bulging symptoms was determined by responding “A little/Moderately/A lot” to the item “Feeling a bulge/lump from or in the vagina” on the Prolapse Quality Of Life (P-QoL) questionnaire. For women with urogenital prolapse, this is a straightforward, valid, and trustworthy questionnaire to gauge the severity of symptoms and how they affect their quality of life [42].

An objective recurrence was defined as a postoperative decline to stage II or below in any compartment based on the POP-Q system or a need for additional surgery. The Subjective Impression of Improvement (PGI-I) score was employed to assess the level of satisfaction following surgery [43]. A seven-point quality-of-life (QoL) measure was used to assess patients’ satisfaction; possible responses range from 1, which indicates “very much improved”, to 7, which indicates “very much worse”. At PGI-I scores ≤ 2, “very much improved” and “much improved” were used to describe QoL success.

The San Gerardo Hospital’s Institutional Review Board in Monza, Italy (SH-MCC 1709/2013), gave its approval for the study. Information was gathered via hospital-specific software for clinical monitoring of patients. One author entered the data into the database, while another author verified it twice. For categorical variables, descriptive statistics were computed as absolute values with percentages, and for continuous variables, the descriptive statistics consisted of the median (interquartile range). For continuous parametric variables, Student’s *t* test was used to examine differences; for continuous non-parametric variables, the Wilcoxon test was used; and for non-continuous variables, the χ^2^ test was used. JMP 7.0 (SAS Institute, Cary, NC, USA) was used for statistical analysis. Significance was defined by a *p* value of less than 0.05.

## 3. Results

In total, 40 patients underwent sacrospinous ligament fixation in the period of interest. Baseline data are shown in Table 1. Most of them (97.5%) were menopausal, with a mean age of 65 years.

Operative data are shown in Table 2. The majority of patients underwent concomitant hysterectomy (92.5%), while three of them (7.5%) preferred a uterine-sparing procedure. Notably, 72.5% of patients underwent anterior repair.

Ten women did not attend the first follow-up visit (mostly patients living out of the county) and were considered lost to follow-up (25%, mostly patients living outside the administrative region). The remaining 30 patients were followed up yearly, with a median follow-up length of 121 [63:131] months. Baseline versus postoperative anatomical comparison demonstrated a significant improvement in all POP-Q points, with a negligible reduction in total vaginal length (Table 3).

However, the rate of anatomical recurrence was remarkably high, as it was observed in 17 (56.7%) patients. Specifically, anterior, central, and posterior recurrences were noted in 53.3%, 13.3%, and 10% of patients. Subjective recurrence was reported by ten (33.3%) women, and reintervention occurred in two (6.7%) patients, while another two (6.7%) women preferred a pessary as a conservative treatment option. From a quality-of-life perspective, according to the PGI-I, twenty-three (76.7%) patients described some degree of improvement after surgery, four (13.3%) described their condition as unmodified, and three (10%) reported some form of worsening after primary treatment.

## 4. Discussion

The surgical treatment of apical prolapse is an intervention that must be chosen based on the type of prolapse and the characteristics and needs of the patient. The suspension of the vaginal vault is of great importance for the stability of the pelvic floor [44,45]. Based on findings from current research, surgeons have favored using the patient’s own fascial and ligament tissue for pelvic floor reconstruction in recent years [46]. DeLancey expounded on this idea by stating that, with the vagina serving as the axis of support, maintaining proper pelvic floor anatomy requires three levels of support. The uterosacral ligament and the cardinal ligament support the cervix and the upper portion of the vagina from the pelvic walls at the first level (apical landmark). The tendinous arch, the pelvic fascia, the fascial arch, and a similar posterior structure are suspended lateral to the middle third of the vagina at the second level (horizontal landmark). At the third level (distal landmark), the vagina is fused with surrounding structures such as the levator ani muscles and the perineal body [47]. Vaginal apical flaws have been described and recommended to be repaired using a variety of techniques. The fact that sacrospinous ligament fixation does not necessitate the installation of mesh has made it frequently employed. Following more than fifty years of research and development, sacrospinous ligament fixation and uterosacral ligament suspension eventually emerged as the standard surgical techniques for treating vaginal vault prolapse [48]. Numerous investigations have contrasted uterosacral ligament suspension (ULS) with sacrospinous ligament fixation (SSLF). In their systematic review and meta-analysis, Yuanzhuo Chen et al. reported evaluating various outcomes related to SSLF and ULS surgery for the treatment of POP [49,50]. The outcomes demonstrated that there were no appreciable variations in surgical success, recurrence, or complications between SSLF and ULS.

However, the posterior approach and location of the sacrospinous ligament (deep and difficult to expose) increase the risk of rectal and nerve injury and postoperative complications such as painful intercourse and pelvic pain [48].

Furthermore, it should be noted that this type of surgery connects the vaginal dome to structures of the pelvis not normally associated with the vagina. The anatomical distortion makes the vagina susceptible to forces and pressures that can later lead to other defects. Specifically, during sacrospinous ligament fixation, the anterior vaginal wall may be subjected to stresses exerted both laterally and superiorly. Theoretically, too much stress in these directions could cause fascial tears, which would cause previously treated defects to relapse or cause new lateral, central, and/or transverse flaws to emerge. The anterior and lateral vaginal walls (left, in the case of the typical fixation of the right sacrospinous ligament) may be subjected to traction enteroceles if they are not correctly maintained, or to pulsion forces that could result in enteroceles [51]. Moreover recurrent prolapse has been reported at sites other than the treated site [52,53]. Colombo and Milani reported recurrence of isolated cystocele (14%) and isolated rectocele (5%) during the 5 years after SSLF performed during vaginal hysterectomy for advanced uterovaginal prolapse.

Our study is the first to evaluate long-term outcomes of sacrospinous ligaments fixation. The data demonstrated that SSL fixation is an effective surgical intervention for prolapse repair, with an improvement in every POP-q parameter. However, we observed anatomical recurrence in more than half patients, with anterior compartment recurrence being particularly common. Moreover, one third of patients experienced recurrence of bulging symptoms, and 13.3% of them required either a pessary or reoperation. Lastly, 10% of patients described an overall worsening of quality of life after surgery.

Our findings suggest that the initial and not completely significant differences in the anterior compartment supports between uterosacral suspension and sacrospinous fixation previously reported by other authors may become clinically relevant in the long term. For instance, Ling-xiao Huang et al., in their study, compared outcomes of uterosacral ligament suspensions and sacrospinous fixation. Following surgery, both groups’ POP-Q scores were much lower than they were prior to the procedure. However, a year following surgery, data revealed that the Aa and Ba values in the USCLF group were lower than those in the SSLF group. The explanation for the variation in Aa and Ba values across the research groups appears to be that the apical vaginal anchors of the USLF are more anterior than those of the SSLF [54,55]. Nonetheless, the results showed that there was no statistically significant difference between the groups’ rates of anterior vaginal prolapse recurrence [56].

Many strategies have been developed for SSLF, such as anterior versus posterior and unilateral versus bilateral fixations, to lessen prolapse recurrence [3]. Proponents of the bilateral technique have led to the unilateral posterior SSLF approach being regarded as the usual procedure despite the vaginal canal’s deviation and the opposite fornix’s weak support [57]. However, in recent years, the anterior approach to sacrospinous fixation has been described and popularized. Differences in the vaginal axis between the posterior and anterior approaches have also been suggested. Parvin et al., in their study, wanted to compare the fixation of the anterior sacrospinous ligament (SSLF) with the standard posterior SSLF to evaluate the differences in terms of complications and surgical outcomes in patients with pelvic organ prolapse (POP) of the apical compartment. Although the anterior approach presents fewer risks and complications than the posterior approach [58], anchoring the vaginal vault to the sacrospinous ligament with anterior access causes a deviation of the vagina such that dyspareunia is created [31]. Consequently, the anterior approach is preferred if repair of posterior compartment prolapse is not indicated [59]. No visceral harm was seen throughout the procedure, according to a retrospective analysis assessing the outcomes of the anterior SSLF technique. Short-term pelvic pain was reported in 6.6% of patients, and recurrences of anterior vaginal wall prolapse and postoperative urine incontinence were noted in 8.3% of patients [18]. Bilateral anterior SSLF was assessed by Solomon et al. for the treatment of apical prolapse. According to their findings, 2.8% of patients experienced extended urine retention in the postoperative phase, and 0.7% of patients needed blood transfusions due to injuries to the bladder or urethra. There was no evidence of nerve damage [59].

To the best of our knowledge, this represents the first study evaluating sacrospinous ligament fixation outcomes after 10 years from the index surgery. Other strengths include the high follow-up rate despite the long-term follow-up and the multimodal evaluation of outcomes, including objective, subjective, and quality-of-life measurements.

Limitations involve the retrospective design and the limited population size. We therefore caution against generalizing our results. Moreover, our study cannot identify risk factors for recurrence due to the small sample size. Further studies with large numbers of cases are needed to identify risk factors for failure after SSLF and evaluate the role of the anterior approach in preventing anterior recurrences.

## 5. Conclusions

In conclusion, transvaginal repair with sacrospinous fixation is a long-lasting option for prolapse primary repair, with an improvement in every POP-q parameter. However, we also showed a considerable proportion of patients reporting a return of bulging symptoms or a general decline in quality of life following surgery, along with a specific grade of anterior recurrences that should be considered.

## Figures and Tables

**Figure 1 healthcare-12-01611-f001:**
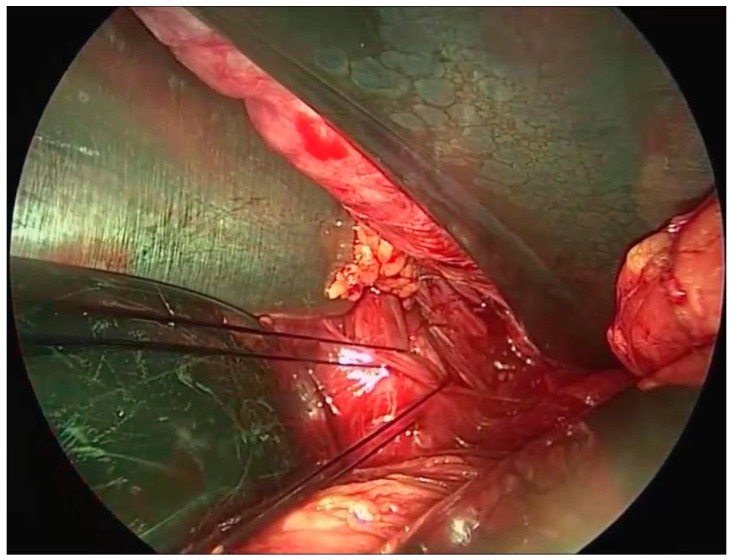
Sacrospinous ligament (SSL) fixation. Image taken with a laparoscope by the transvaginal route for didactical purposes.

**Table 1 healthcare-12-01611-t001:** Population baseline characteristics. Continuous data are presented as median [interquartile range]. Non-continuous data are presented as absolute (relative) frequency.

Variable	Result
Age (years)	65 [60:70]
Menopausal status	39 (97.5%)
Parity (n)	2 [2:3]
Birth weight of largest baby (g)	3500 [3100:3950]

**Table 2 healthcare-12-01611-t002:** Operative data. Continuous data are presented as median [interquartile range]. Non-continuous data are presented as absolute (relative) frequency.

Variable	Result
Sacrospinous fixation	40 (100%)
Hysterectomy	37 (92.5%)
Hysteropexy	3 (7.5%)
Anterior repair	29 (72.5%)
Posterior repair	23 (57.5%)
Operative time (min)	99 [82:115]
Blood loss (mL)	200 [100:250]

**Table 3 healthcare-12-01611-t003:** POP-Q data. Continuous data are presented as median [interquartile range].

	Baseline	Postoperative	*p*-Value
Aa	2.0 [−0.3:3.0]	−1.0 [−2.0:0]	<0.001
Ba	2.5 [−0.3:4.0]	−1.0 [−2.0:0]	<0.001
C	2.0 [−3.0:6.0]	−6.0 [−3.0:−7.0]	<0.001
gh	5.0 [3.8:5.0]	3.0 [3.0:3.6]	<0.001
pb	2.0 [1.8:2.0]	3.0 [3.0:3.0]	<0.001
tvl	9.0 [9.0:10.0]	9.0 [8.0:10.0]	0.006
Ap	−1.0 [−1.0: 2.0]	−2.0 [−3.0:−2.0]	<0.001
Bp	−1.0 [−1.0: 2.0]	−2.0 [−3.0:−2.0]	<0.001

## Data Availability

The data presented in this study are available on request from the corresponding author.

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
