# Peer review of "Long-Term Outcomes (10 Years) of Sacrospinous Ligament Fixation for Pelvic Organ Prolapse Repair"

_healthcare, 2024, doi:10.3390/healthcare12161611_

Round 1
Reviewer 1 Report
Comments and Suggestions for Authors
Dear Authors,
The manuscript entitled "Long Term Outcomes (10 Years) of Sacrospinous Ligaments Fixation for Pelvic Organ Prolapse Repair" describes results of Sacrospinous ligament fixation. The work explains significant details of the issue. However, authors need to address following issues.
1. Introduction contains too much of information than needed. Kindly rewrite in concise way.
2. Sample considered is very small. Therefore, the outcomes of the result could not be generalized as provided in discussion.
3. Authors need to provide some more inputs related to work as the sample is very small. This may increase significance of the work carried out.
4. In addition to small sample size, results could not be summarized in general for the sample considered initially as some the patients have shown up to follow up treatment. Hence, percentage outcome in each case much be recalculated.
5. finally, revise the manuscript as suggested above and rewrite the conclusion accordingly.
Comments on the Quality of English LanguageQuality of English language of the manuscript is fine except for minor mistakes.
Author Response
Dear Reviewer 1,
The manuscript entitled "Long Term Outcomes (10 Years) of Sacrospinous Ligaments Fixation for Pelvic Organ Prolapse Repair" describes results of Sacrospinous ligament fixation. The work explains significant details of the issue. However, authors need to address following issues.
- Introduction contains too much of information than needed. Kindly rewrite in concise way.
Author's reply: Edited
- Sample considered is very small. Therefore, the outcomes of the result could not be generalized as provided in discussion.
Author's reply: Edited and clearly added as limitation. “Limitations involve the retrospective design and the limited population size. Thus, our results may not be generalizable.”
- Authors need to provide some more inputs related to work as the sample is very small. This may increase significance of the work carried out.
Author's reply: This represent the first long term outcomes study, and given the long follow-up, a small population is currently anovoidable; However this is clearly stated as limitation
- In addition to small sample size, results could not be summarized in general for the sample considered initially as some the patients have shown up to follow up treatment. Hence, percentage outcome in each case much be recalculated.
Author's reply: We respectfully disagree. All studies has a loss at follow-up. This has a 25% loss at follow up, which is really expected considering the benign disease and the long term follow-up, and comparable to similar studies
- finally, revise the manuscript as suggested above and rewrite the conclusion accordingly.
Author's reply: Conclusion edited
Reviewer 2 Report
Comments and Suggestions for Authors
the Authors present the results of a long-term analysis of sacrospinous ligament fixation in patients with vaginal prolapse. The study is well conducted and has an impressive follow-up.
Just one major comment: I don't undestand how the POP-Q (Italian version published in 2022) could be applied to a retrospectively selected population. am I missing something?
minor: the Introduction is quite long, and should be shortened.
Comments on the Quality of English Languagefine, only some polishing will be required at the proof-reading stage.
Author Response
Dear Reviewer 2,
Comment: The Authors present the results of a long-term analysis of sacrospinous ligament fixation in patients with vaginal prolapse. The study is well conducted and has an impressive follow-up.
Author's reply: Thank you very much!
Comment: Just one major comment: I don't undestand how the POP-Q (Italian version published in 2022) could be applied to a retrospectively selected population. am I missing something?
Author's reply: The POP-q classification system is available since 1996; we added details for clarity.
Comment: minor: the Introduction is quite long, and should be shortened.
Author's reply: Edited
Reviewer 3 Report
Comments and Suggestions for Authors
A very interesting article was sent for my review. This study aimed to evaluate the 10-year results of sacrospinous ligament suspension as primary repair for apical prolapse and to evaluate long-term side effects.
The article is written correctly, but I have a few comments:
1. In Table 1, the authors presented the characteristics of the study group, but the record is quite chaotic. I suggest the authors carefully explain in the table description what the entry [2: 3], [60-70] means. I believe that the authors should standardize this provision.
2. The tables are missing headers: variable/result
3. I suggest the authors explain in more detail what the Pelvic Organ Prolapse Quantification (POP-Q) System means. And how this system works, how many points it contains, and what it means. Otherwise, for people less familiar with the topic, table 3 will be completely unreadable.
2. The Italian Prolapse Quality of Life (P-QoL) - what is this questionnaire based on? The same is for The Subjective Impression of Improvement (PGI-I) score. They should be at least explained in the text. What do they consider, and how do the patients receive the points?
2. A study design/scheme figure can be helpful. It would allow for a better understanding of what was assessed and when.
3. The discussion is correct; the authors correctly refer to the obtained results and comment on them appropriately.
4. When including patients in the study, did the authors consider the coexistence of additional diseases that could influence the results?
5. The text has minor linguistic errors; I suggest you reread the article and spot them.
E.g. "Conclusion: Ttransvaginal.."
Minor editing of the language is required.
Author Response
Dear Reviewer 3,
Comment: The article is written correctly, but I have a few comments:
- In Table 1, the authors presented the characteristics of the study group, but the record is quite chaotic. I suggest the authors carefully explain in the table description what the entry [2: 3], [60-70] means. I believe that the authors should standardize this provision.
Author's reply: Edited
- The tables are missing headers: variable/result
Author's reply: Edited
- I suggest the authors explain in more detail what the Pelvic Organ Prolapse Quantification (POP-Q) System means. And how this system works, how many points it contains, and what it means. Otherwise, for people less familiar with the topic, table 3 will be completely unreadable.
Author's reply: Thank you for the suggestion, added
- The Italian Prolapse Quality of Life (P-QoL) - what is this questionnaire based on? The same is for The Subjective Impression of Improvement (PGI-I) score. They should be at least explained in the text. What do they consider, and how do the patients receive the points?
Author's reply: These are self evaluation tools explained in materials and methods section
The subjective recurrence of postoperative bulging symptoms was determined by responding “A little/Moderately/A lot” to the item “Feeling a bulge/lump from or in the vagina” on the Prolapse Quality Of Life (P‐QoL) questionnaire. This is a simple, valid, reliable questionnaire to assess the severity of symptoms and their impact on the quality of life in women with urogenital prolapse [Ref]
The Subjective Impression of Improvement (PGI-I) score was employed to assess the level of satisfaction following surgery. [Ref] A seven-point quality of life (QoL) measure is used to assess patients' satisfaction; responses range from 1, which indicates "very much improved," to 7, which indicates "very much worse." At PGI-I score (≤2), "very much improved" and "much improved" were used to describe QoL success.
- A study design/scheme figure can be helpful. It would allow for a better understanding of what was assessed and when.
Author's reply: Tools are well explained in the materials and methods section. However references were provided for those who want go deeper
- The discussion is correct; the authors correctly refer to the obtained results and comment on them appropriately.
Author's reply: thanks for the comment
- When including patients in the study, did the authors consider the coexistence of additional diseases that could influence the results?
Author's reply: Thanks for the observation. No, this type of analysis was not carried out in consideration of the limited population size; this was acknowledged as limitation in the proper section
- The text has minor linguistic errors; I suggest you reread the article and spot them.E.g. "Conclusion: Ttransvaginal.."
Author's reply: thanks for the comment, edited
Round 2
Reviewer 1 Report
Comments and Suggestions for Authors
Dear Authors,
The manuscript has been revised enough to be published.